# The negligible role of carbon offsetting in corporate climate strategies

Niklas Stolz [1] ✉ & Benedict S. Probst [1,2,3]

Carbon credits feature prominently in corporate climate strategies and have sparked public debate about their potential to delay companies' internal decarbonisation. While industry reports claim that credit purchasers decarbonise faster, rigorous evidence is missing. Here, we provide an in-depth analysis of 89 multinational companies' historical emission reductions and climate target ambitions. Based on self-reported environmental data and more than 400 sustainability reports, we find no significant difference between the climate strategies of companies that purchased credits and those that did not. Voluntary offsetting is not a central part of most companies' climate strategies, and many pass credit costs directly onto their customers. While the companies within our sample retired one-fourth of all carbon credits in 2022, the top five offsetters' expenditures on voluntary emission offsetting are, on average, only 1 percent relative to their capital expenditures. For most companies, carbon credits are, therefore, unlikely to crowd out internal decarbonisation measures. Yet, we document that for large-scale offsetters in the airline industry, carbon credit purchases competed with financing internal decarbonisation measures.

Companies have faced increasing pressure from policymakers, consumers, civil society, and investors to reduce emissions. In response to the Paris Agreement, governments have consistently tightened their climate policies[1]. This regulatory shift has increased investors' concerns about transition risk, prompting them to demand credible climate strategies from the companies in their portfolio[2]. However, media and non-governmental organisations (NGOs) that monitor corporate emission targets[3] frequently expose cases of greenwashing[4,5].

In response to the growing pressure, half of the Forbes Global 2000 companies have set net zero targets[3]. However, no globally accepted, binding standard for setting credible net zero targets exists. This gap has been filled by private initiatives like the Science-Based Targets initiative (SBTi), which provides standards on the emission coverage and ambition of net zero targets. SBTi also offers guidelines for how companies can compensate for residual emissions once they have reduced most of their emissions[6].

Many companies use or plan to use carbon credits from the voluntary carbon markets to offset parts of their emissions on their path to net zero[7]. Carbon credits are the "reduction, avoidance or removal of a unit of greenhouse gas (GHG) emissions by one entity, purchased by another entity to counterbalance a unit of GHG emission by that other entity"[8]. These voluntary emission practices and the claims derived from emission offsetting (e.g. carbon neutral, net zero)[9] are prominently featured in the public debate about decarbonisation[10–12]. However, carbon credits' role in corporate climate strategies remains unclear.

Insights from political economy, stakeholder theory, and legitimacy theory suggest that when civil society, politicians, and other stakeholders pressure companies, they may engage in voluntary social or environmental disclosure to shore up support[13,14]. However, these strands of literature typically do not predict or assess whether such action translates into improved corporate climate performance. Companies may engage in voluntary environmental disclosure to

[1]Group for Sustainability and Technology, ETH Zurich, 8092 Zurich, Switzerland. [2]Net Zero Lab, Max Planck Institute for Innovation and Competition, Munich, Germany. [3]Cambridge Centre for Environmental, Energy and Natural Resource Governance, University of Cambridge, Cambridge, UK. ✉e-mail: nstolz@ethz.ch

achieve a variety of objectives: they may aim to inform the public about superior environmental performance relative to peers, to reshape the company's image without substantial change, to shift the focus of the discourse, or to influence stakeholder expectations about the company's actions[13,15]. In the context of voluntary carbon offsetting, companies may retire carbon credits to signal superior environmental performance or enhance their public image without substantially advancing internal decarbonisation. Alternatively, corporate emission offsetting, as a form of corporate social responsibility, may be driven by altruistic motives[16] that have not been previously captured in the literature.

Purchasing carbon credits instead of pursuing potentially more effective internal decarbonisation can be conceptualised as a moral hazard. Moral hazard occurs when actors take on higher risks or engage in socially suboptimal behaviour because they are shielded from the consequences of their actions[17]. Therefore, there is a risk that emission offsetting leads to moral hazard when companies neglect internal and value chain emission reductions because the improved public perception achieved through emission offsetting shields them from the risk of reputational damage, public scrutiny, or governmental regulation.

There is mixed evidence regarding the relationship between corporate carbon management practices and subsequent emission reductions. While some studies demonstrate a positive relationship between corporate carbon disclosure and emission reductions[18,19], others find this link only among emission-intensive companies[20]. Conversely, to our knowledge, no study to date has established a significant relationship between adopting reporting guidelines, such as the Global Reporting Initiative (GRI) and improved corporate emission performance[21,22]. Further, the impact of corporate climate strategies on emission reductions remains ambiguous. For example, there is limited evidence around the relationship between the mere presence of corporate climate targets and subsequent decarbonisation[23], though more ambitious targets are associated with greater emission reductions[23,24]. Recent findings suggest that only a comprehensive mix of corporate climate instruments (e.g., absolute emission targets, internal carbon prices, value chain engagement) is linked to absolute emission reductions[25].

While research on companies' use of carbon credits is still nascent, research on renewable energy attributes (REAs), another market-based carbon accounting tool, is more advanced. REAs allow companies to verify and claim the purchase of renewable energy, directly reducing market-based scope 2 emissions under the Greenhouse Gas Protocol and Science-Based Targets initiative (SBTi)[26]. Unlike voluntary carbon credits, REAs can be counted towards SBTi goals. Ascui et al.[27] show that companies using REAs tend to increase their scope 1 and 2 emissions without improving energy efficiency compared to peers who do not use them, which indicates their potential to induce moral hazard[27]. Additionally, setting targets for market-based scope 2 emissions and achieving them by purchasing REAs might undermine the integrity of SBTi because these certificates do not lead to real emission reductions[28–30].

In contrast to the well-documented role of REAs in corporate climate performance, research on carbon credits is still nascent. Recent industry reports claim that companies engaging in emission offsetting decarbonise faster than their peers[31–33]. These industry reports argue that by voluntarily offsetting emissions, companies put a price on greenhouse gas emissions, creating a financial incentive for faster internal decarbonisation[33]. In addition, Engler et al.[34] show that emission offsetting does not crowd out investment in decarbonisation for German small- to medium-sized companies but complements pro-environmental activities[34].

However, growing evidence challenges voluntary carbon markets' overall positive climate effect. A range of studies show that emission reductions associated with carbon credits are systematically overestimated[35]. Also, large-scale offsetters tend to source cheap and low-quality carbon credits[36]. Beyond these issues, the industry reports supporting the claim that emission offsetters decarbonise faster than their peers do not control for company size and do not examine whether carbon credit spending might crowd out investments in internal decarbonisation[31–33]. Further, scientific findings on the corporate usage of carbon credits are not based on observed corporate emission performance but capture explanatory variables like perceived climate-related risks, presence of any climate target and environmental management systems[34]. However, these findings cannot be generalised to multinational companies in hard-to-abate sectors, which constitute the largest buyer group[37].

Here, we analyse the emission offsetting behaviour of 89 multinational companies in the oil and gas (O&G), automobile manufacturing, and airline sectors. The companies in the sample constitute 24% of carbon credits retired in the voluntary carbon market in 2022. We focus on multinational companies in hard-to-decarbonise sectors since those companies are characterised by high emissions, low profits per ton of emitted carbon dioxide[38], and weak climate targets[37]. We compiled a dataset on the corporate usage of voluntary carbon credits by combining several sources. First, we utilised data from CDP on corporate greenhouse gas emissions, carbon credit retirements, and emission targets. Second, we cross-verified the carbon credit retirement data with the largest registries (Verra, Gold Standard, CDM). Third, we compiled a dataset of qualitative information on corporate emission offsetting between 2014 and 2023, analysing over 400 corporate sustainability and annual reports. Our analysis proceeds in two stages. First, we evaluate if emission offsetting is associated with more ambitious decarbonisation efforts by companies compared to peers that do not engage in emission offsetting. We measure decarbonisation ambition with two metrics: the change in scope 1 emissions between the CDP reporting cycles 2018 and 2023 as a retrospective indicator and the ambition of emission reduction targets as a forward-looking metric. Second, we examine the role of emission offsetting in corporate climate strategies by analysing carbon credit expenditures and their usage details.

We find no significant difference in climate strategy between companies that offset emissions and those that do not. The reason for the non-significant relationship is likely the low expenditures on emission offsetting relative to companies' capital expenditures, the low overall share of offset emissions relative to total company emissions, and the non-binding nature of voluntary emission offsetting. However, we find that carbon credit expenditures compete with financing internal decarbonisation efforts for large-scale offsetting campaigns. We argue that voluntary emission offsetting is not associated with accelerated corporate decarbonisation and that the role of carbon credits in corporate climate strategies is overstated in the public discourse.

## Results

### Emission offsetting is not associated with better environmental performance

We find no significant relationship between the number of carbon credits a company retired in CDP's 2023 reporting period and the historical change in scope 1 emissions or with the ambition of their emission targets (see Fig. 1).

The only significant association with the change of scope 1 emissions over the study period is that companies in the oil and gas sector reduced their emissions more slowly than companies in the aviation sector (significance level $\alpha = 0.05$, Fig. 1a). However, this relationship is not robust to leaving single observations out of the regression (see Supplementary Information Fig. S1a, b). Nevertheless, there is no significant association between the number of carbon credits retired and the change in scope 1 emissions.

## (a)

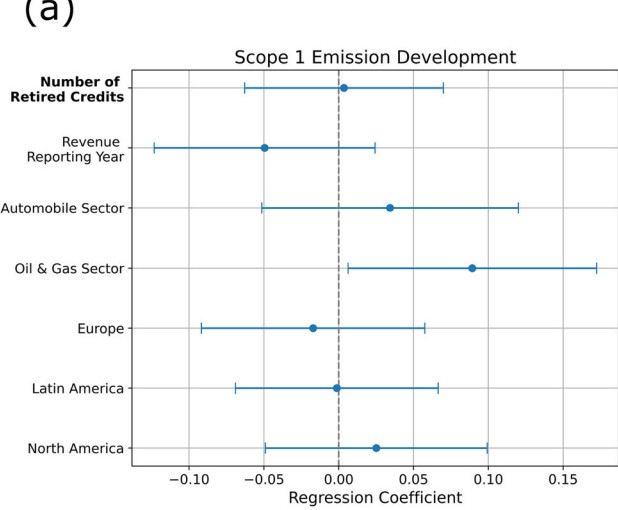

## (b)

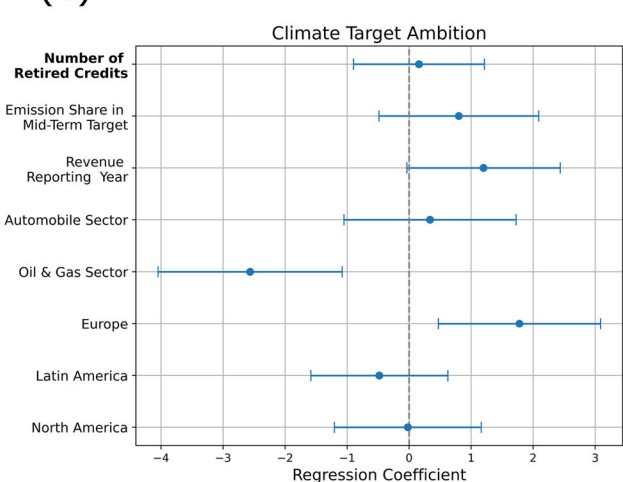

**Fig. 1 | Ordinary Least Squares (OLS) regression coefficients for (a) scope 1 emissions ratio (CDP 2023/2018) with $n = 78$ (b) and climate target ambition with $n = 89$.** The graph displays the estimated regression coefficients ($\hat{\beta}_i$), with error bars representing their 95% confidence intervals (CI) with $CI_{\beta_i, 0.95} = \left[ \hat{\beta}_i - t^* \cdot SE(\hat{\beta}_i), \hat{\beta}_i + t^* \cdot SE(\hat{\beta}_i) \right]$ and $t^*$ the critical value from the t-distribution. In (a), positive regression coefficients indicate a negative relationship between the explanatory variables (on $y$-axis) and decarbonisation speed ($x$-axis), suggesting that as the explanatory variables increase, we observe a decreased

decarbonisation speed. In (b), positive regression coefficients indicate a positive relationship between the explanatory variables ($y$-axis) and climate target ambition ($x$-axis), suggesting that as the explanatory variables increase, we observe an increased climate target ambition. The sectoral categorical variables are relative to the aviation sector, and the geographic categorical variables are relative to headquarters in Asia. The label of retired carbon credits is written in bold as it represents the study's primary outcome variable of interest.

Conversely, the variation in climate target ambition until 2050 is significantly correlated with sector and location. Companies headquartered in Europe set more ambitious climate targets than their peers in other regions. Conversely, oil and gas companies are associated with setting targets that are less ambitious than those of companies in other sectors. However, we find no significant association between the number of retired carbon credits and climate target ambition.

These findings suggest that the number of carbon credits a company retires is unrelated to its past emission reductions and climate target ambition. This conclusion holds across various robustness checks (see Supplementary Information Tables S3–S8). Below, we explore the reasons for these findings.

## Most companies spend negligible funds on carbon credits

There are two opposing lines of argument about how the purchase of carbon credits can influence companies' emission trajectories. Some argue that buying carbon credits diverts resources from internal decarbonisation, thereby delaying it[39]. Others argue that the cost of carbon credits is a voluntary penalty for greenhouse gas emissions, which incentivises companies to accelerate decarbonisation[33]. Our findings support none of these views since we find a non-significant relationship between offsetting and emission trajectories and climate target ambition.

To delve deeper into potential reasons behind our findings, we estimate the magnitude of emission offsetting costs relative to companies' investments in their assets and mandatory emission costs. Therefore, we estimate companies' carbon credit and emission trading scheme costs and compare them to their reported capital expenditures. Compared to their capital expenditures (CAPEX), companies allocate relatively small amounts of funds to emission offsetting (see Fig. 2). The companies with the largest CAPEX share spent on carbon credits are easyJet (2.7%) and Delta Air Lines (1.8%). In the oil and gas sector, Eni (0.14 −0.38%) and Shell (0.10 −0.25%) follow. In the automobile manufacturing sector, Volkswagen Group (0.13−0.20%) and Mercedes-Benz Group (0.07 −0.13%) spend the largest share of funds on offsetting. Overall, the distribution of corporate spending on

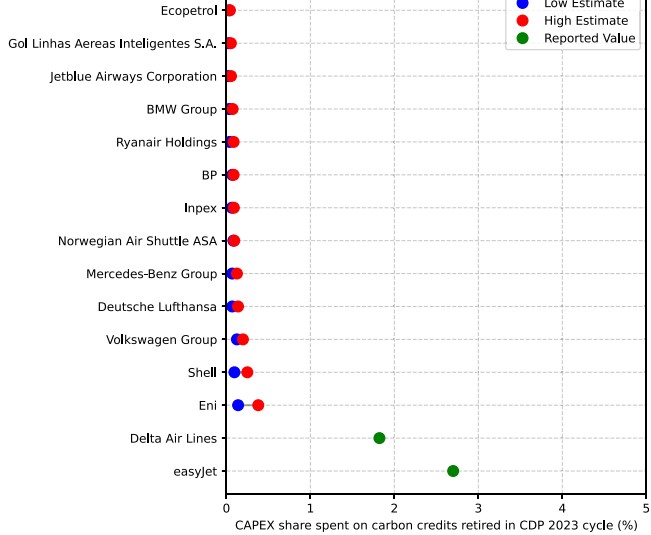

**Fig. 2 | Costs of purchasing carbon credits for voluntary emission offsetting relative to companies' capital expenditure (CAPEX) during the reported year.** Minimum estimates (blue) are based on the lowest reported carbon credit price among companies in the sample (easyJet) for 2022, and maximum estimates (red) are based on data from Ecosystem Marketplace (2023)[60]. Reported CAPEX shares (green) indicate that companies reported total spending for carbon credits in 2022. The figure includes the 15 companies with the highest share of funds spent on carbon credits relative to their CAPEX.

emission offsetting is long-tailed, meaning that few companies spend a lot on credits while the majority spend little.

Similarly, European companies face much higher costs under compliance Emission Trading Schemes (ETS) - such as EU-ETS, UK-ETS, and Switzerland-ETS - compared to their spending on carbon credits (see Fig. 3). However, there is substantial variation between sectors. For example, easyJet, despite being an outlier in carbon credit spending, spent 9.9 times more money on compliance emission

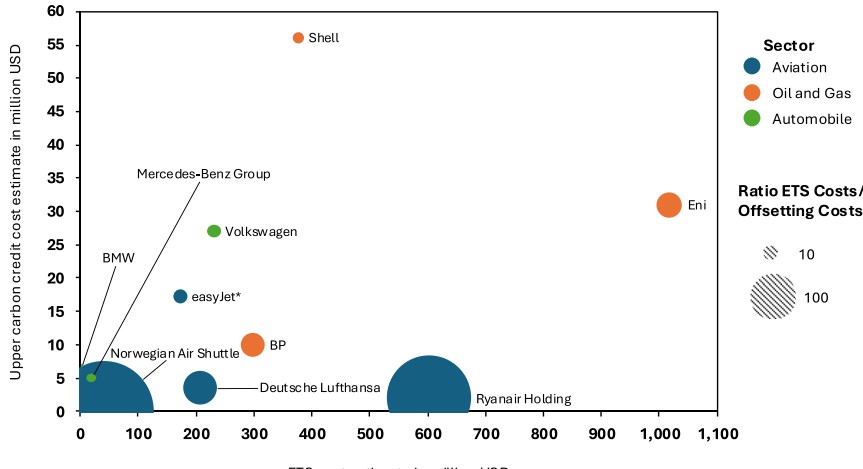

**Fig. 3 | Upper estimate of funds that European companies spent on carbon credits (reported in CDP's 2023 survey) compared to estimated spending on European Emission Trading Scheme (ETS) allowances.** Sources: Retired carbon credits and ETS allowances from the CDP database[40], average ETS prices from World Bank[46], Carbon credit price estimates from Ecosystem Marketplace (2023)[60]. *Easyjet directly reports spending on carbon credits in their 2022 annual report.

trading than on voluntary emission offsetting. In the oil and gas sector, Eni had to allocate ~33.1 times more to compliance emission trading, while Shell only spent ~6.7 times more. An exception to this trend is the BMW Group, which spent more on voluntary offsetting - 2.3 times their compliance emission trading costs - since they could cover their emissions with excess allowances from past years[40].

### The role of emission offsetting in corporate climate strategies is volatile

In addition to the financial commitment of offsetting emissions, which emissions a company offsets, who pays for the carbon credits, and the permanence of the decision to offset emissions likely affect the role of carbon credits in companies' decarbonisation strategies. We find that the role of emission offsetting in corporate climate strategies varies between companies (see Fig. 4). Companies that purchase large quantities of carbon credits typically follow one of two approaches: they either offset a fixed portion of their emissions (e.g. Delta Air Lines, easyJet, and Volkswagen Group) or offset residual emissions to meet specific emission targets (e.g. Shell, Eni, and Inpex). In contrast, companies that purchase fewer carbon credits typically only offer them to their customers during the checkout (e.g. Deutsche Lufthansa, Air France - KLM, Ryanair Holding, International Consolidated Airlines Group) or offset their business travel (e.g. Deutsche Lufthansa, International Consolidated Airlines Group).

We find that all airlines and oil and gas companies that reported the retirement of >100,000 carbon credits in the CDP 2023 survey either provide customers the option to voluntarily offset their purchases at checkout or to buy pre-offset fossil products. It is common practice for airlines to offer the option to offset emissions during the flight booking process (e.g. easyJet post-2022, Deutsche Lufthansa, International Consolidated Airlines, Ryanair Holdings, Air France - KLM). Similarly, oil and gas companies regularly sell pre-offset fossil products (e.g. Shell, BP, Eni, Inpex, TotalEnergies). Therefore, despite their relatively low expenditure on carbon credits, many companies pass these credits' costs and purchase decisions directly onto their customers.

Besides large variability in the usage of carbon credits for emission offsetting, companies regularly change their offsetting strategy. Over the past decade, many companies have expanded the use cases for emission offsetting. However, the largest two emission offsetters, Delta Air Lines and easyJet, stopped their large-scale offsetting campaign in 2022. Also, BP changed its offsetting strategy in

2020, ceasing to use carbon credits to reach its zero net growth target[41].

### Most companies offset a small share of their total emissions
Given the large variety in companies' approaches to emission offsetting (Fig. 4), it is unclear how substantial the number of retired carbon credits is compared to companies' overall greenhouse gas emissions. We find that only easyJet (78.2%) and Delta Air Lines (43.7%) retire carbon credits accounting for substantial portions of their total scope 1, 2, and 3 emissions (see Fig. 5a). Deutsche Lufthansa offsets the third highest share of scope 1, 2 and 3 emissions (1.5%) despite ranking 11th in the number of retired carbon credits in the sample and only offsetting employee business travel and offering voluntary offsets to customers (see Fig. 4). Among oil and gas companies, Eni (1.4%) and BP (0.7%) offset the largest shares of their emissions, while in the automobile manufacturing sector, Volkswagen Group (1.1%) and BMW (0.7%) offset the most.

The picture changes substantially when focusing solely on scope 1 and location-based scope 2 emissions. The three automobile manufacturers that offset their emissions, Volkswagen Group (50.3%), BMW (45.2%), and Mercedes-Benz Group (40.3%), emerge among the largest five offsetters of scope 1 and 2 emissions (see Fig. 5b). These offset shares rise further when considering market-based scope 2 emissions since BMW and Mercedes-Benz Group offset their full scope 1 and market-based scope 2 emissions. However, even limited to scope 1 and 2 emissions, oil and gas companies offset minor emission shares. Shell (9.8%) has the largest share of offset scope 1 and 2 emissions in the oil and gas sector, followed by Eni (7.5%) and BP (7%).

### Large-scale offsetting might compete with internal decarbonisation
Despite clear indications that voluntary emission offsetting contributes, on average, little to meaningful decarbonisation efforts (see Fig. 1) due to low investment levels (see Fig. 2) and low persistence of emission offsetting in corporate climate strategies (see Fig. 4), it is unclear if voluntary emission offsetting competes with internal decarbonisation investment in some cases. Among companies engaging in emission offsetting, we identify two key pathways through which offsetting may compete with internal decarbonisation efforts.

The first pathway is the investment effect. In this pathway, companies allocate a fixed budget for decarbonisation, which includes both the purchase of carbon credits and investments in internal

| Company Name | Retired Credits CDP 2023 (mln.) | 2014 | 2015 | 2016 | 2017 | 2018 | 2019 | 2020 | 2021 | 2022 | 2023 | Notes |
|---|---|---|---|---|---|---|---|---|---|---|---|---|
| Delta Air Lines | 18.9 | | | | | | | | | | X | Delta Air Lines stopped offsetting all emissions on 31 March 2022. |
| easyJet | 6.3 | X | X | X | X | X | | | | | | easyJet stopped offsetting all scope 1 & 2 emissions on 31 December 2022. |
| Shell | 5.8 | X | X | X | X | X | | | | | | Emission offsetting is included in net carbon intensity goal, also when purchased by customer. |
| Volkswagen Group | 4.6 | X | X | X | X | X | | | | | | Offsets supply chain emissions of electric vehicles. |
| Eni | 3 | X | X | X | X | X | X | | | | | Emission offsetting is included in net carbon intensity goal, also when purchased by customer. |
| BP | 2.3 | | | | | | | | | | | From 2017 - 2019 carbon credits were allowed in no net emission growth target. |
| BMW Group | 0.85 | X | X | X | X | X | X | X | | | | Offset scope 1 & 2 emissions - changed narrative from "neutralising" emissions to "beyond value chain mitigation" in 2023. |
| Mercedes-Benz Group | 0.68 | X | X | X | X | X | X | X | X | | | Offsets scope 1 & 2 emissions of production plants. |
| TotalEnergies | 0.64 | X | X | X | X | | | | | | | Plans to start offsetting residual emissions starting in 2030. |
| Deutsche Lufthansa | 0.5 | | | | | | | | | | | Has offered offsetting program to customers since 2007. |
| Ecopetrol | 0.42 | X | X | X | X | X | X | X | | | | Subsidiaries in midstream are certified as carbon neutral. |
| OMV Group | 0.26 | X | X | X | X | X | | | | | | Offers customers opportunity to offset their emissions. |
| Inpex | 0.25 | | | X | X | X | X | X | | | | Emission offsetting is part of net carbon intensity target. |
| International Consolidated Airlines Group | 0.23 | | | | | | | | | | | Until 2020 customers could donate to British Airways Carbon fund. |
| Ryanair Holdings | 0.2 | X | X | X | X | | | | | | | Offers customers opportunity to offset their emissions. |
| Air France - KLM | 0.1 | | | | | | | | | | | Air France offsets domestic flights since 2020, which became mandatory in France 2022. |

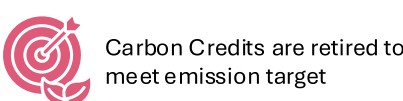
Carbon Credits are retired to meet emission target

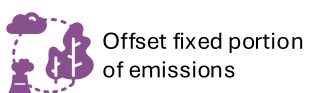
Offset fixed portion of emissions

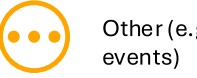
Other (e.g. one-time events)

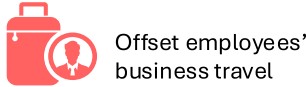
Offset employees' business travel

Offering customers to offset purchases or offer pre-offset fossil products

**Fig. 4 | Emission sources offset by companies.** Overview of offsetting purpose for companies that reported >100,000 retired credits in CDP's 2023 survey between 2014 and 2023. The information is based on companies' sustainability and annual reports.

emission reduction measures. Every dollar spent on carbon credits reduces the funds available for investments in operations or value chain decarbonisation. As a result, carbon credit purchases compete with investments in operations or value chain decarbonisation, potentially replacing internal decarbonisation efforts.

The second pathway is the target effect. Here, companies use carbon credits as part of their strategy to meet emission reduction targets. Since purchasing carbon credits is often cheaper and easier than implementing structural changes, companies may opt to purchase carbon credits instead of implementing more substantial

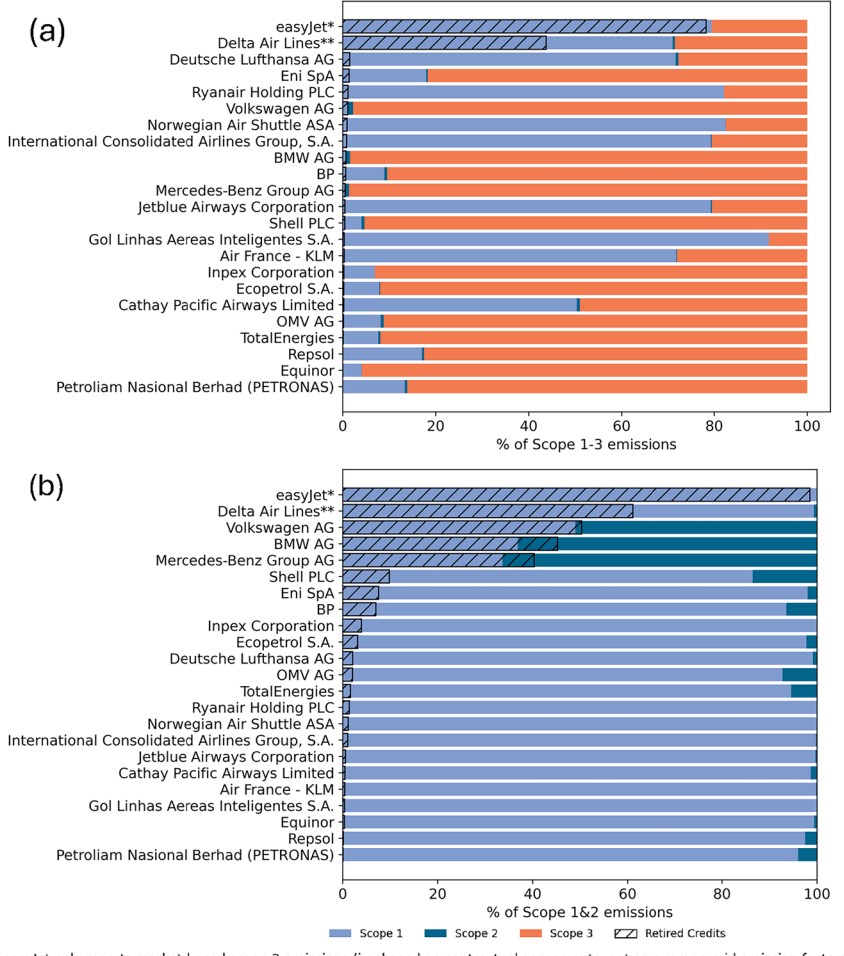

**Fig. 5 | Emission share that companies voluntarily offset.** Share of scope 1, 2, and 3 emissions (**a**) and of scope 1 and 2 emissions (**b**) that companies voluntarily offset based on emission reporting during the 2023 CDP reporting cycle[40]. Location-based scope 2 emissions are used.

internal decarbonisation initiatives to meet their targets. Therefore, as for the investment effect, the target effect might indirectly lead to competition between investments in internal decarbonisation and carbon credit expenditures.

We find evidence for the investment effect, where investments in decarbonisation are crowded out due to the allocation of funds to carbon offsetting for Delta Air Lines and easyJet. These two companies are the outliers, spending the largest CAPEX ratios on carbon credits (see Fig. 2). Delta Air Lines set a fixed budget of USD 1 billion over 10 years for decarbonisation measures, including both carbon credits and investments in internal decarbonisation initiatives[42]. After 3 years, Delta spent USD 284 million on carbon credits, leaving only USD 16 million for internal decarbonisation initiatives, assuming an annual budget of USD 100 million over 10 years (see Table 1). Although easyJet did not communicate a fixed decarbonisation budget, after stopping their large-scale offsetting campaign, they announced that they would use the funds formerly committed to emission offsetting for internal decarbonisation measures, indicating that large-scale offsetting competed with internal decarbonisation investments.

In the oil and gas sectors, we observe the target effect, where emission offsetting is treated as a substitute for emission reductions. Shell, Eni, and Inpex include carbon credits in their emission reduction targets (see Table 1), allowing them to meet their targets either by reducing emissions or by retiring carbon credits. While Eni communicates how many carbon credits they plan to use to achieve their

emission targets[43], it is unclear how many carbon credits Shell[44] and Inpex[45] plan to use for achieving their emission targets.

## Discussion
Our analysis of 89 multinational companies, covering around one-fourth of all carbon credits retired in 2022, reveals no significant difference in climate performance or ambition between companies that offset their emissions and those that do not. This conclusion holds for historical emission performance (CDP 2018-2023 survey) and forward-looking climate target ambition. Contrary to the findings by industry reports[31–33] and Engler et al. (2023)[34], companies in hard-to-decarbonise sectors appear to purchase and retire carbon credits primarily in response to external pressure. Given that most companies in our sample spend little and inconsistently on carbon credits, it appears that carbon credit purchases are a strategy to maintain or restore legitimacy without accelerating decarbonisation relative to their peers.

Compared to their overall capital expenditures, companies allocate very little funds to emission offsetting (Fig. 2). Even the two outliers, easyJet and Delta Air Lines, spend less than 3% relative to their CAPEX on carbon credits. Also, compared to compliance emission trading schemes, the costs of voluntary carbon offsetting are relatively low where such schemes exist. For instance, the five European companies that retired the most carbon credits spent, on average, 5.7% of their compliance carbon expenditures on voluntary carbon credits. Therefore, even reallocating these funds to internal decarbonisation

**Table 1 | Evidence of competition between voluntary emission offsetting and internal decarbonisation**

| Company | Delay Pathway | Case |
|---|---|---|
| Delta Air Lines | Investment effect | In February 2020, Delta Air Lines committed to spend USD 1 billion towards climate neutrality over 10 years. This budget is shared between carbon reductions within Delta, scaling of sustainable aviation fuel, advancing new technologies, and expanding their offset portfolio[42]. |
| easyJet | Investment effect | After stopping large-scale offsetting, easyJet announced that they would "transition [their] investments from out-of-sector carbon offsetting into supporting [...] the individual elements of [their] roadmap, to decarbonise [their] operations."[64] |
| Shell | Target effect | Shell's scope 3 targets for 2030 and 2050 based on net carbon intensity allows the use of carbon credits to offset emissions[44]. |
| Eni | Target effect | Decarbonisation targets for scope 1,2 and 3 allow carbon credits ("Net Carbon Footprint" and "Net GHG Lifecycle Emissions")[43]. |
| Inpex | Target effect | Inpex's scope 1 and 2 target for 2030 allows for emission offsetting[45]. |

Evidence of competition between voluntary emission offsetting and internal decarbonisation in corporate annual and sustainability reports.

efforts would likely have minimal impact on overall emission reductions. This illustrates that despite its prominence in the public debate on corporate climate strategies, voluntary emission offsetting plays only a minor role in shaping these strategies. Hence, its importance is overstated in public discourse since it imposes minimal financial burden on companies and requires little commitment to climate change mitigation since companies can cease these activities at any time.

Conversely, companies operating under European emission trading schemes already pay substantial costs for their emissions. Unlike voluntary carbon markets, these compliance schemes are mandatory, and carbon allowances trade at substantially higher prices than carbon credits on the voluntary carbon markets[46]. Consequently, they are more suitable for making a business case for corporate decarbonisation.

Although, on average, there is no statistically significant association between voluntary emission offsetting and companies' environmental performance, qualitative evidence suggests two potential ways emission offsetting could compete with internal decarbonisation. The first is the investment effect, which arises when companies allocate a shared decarbonisation budget for both carbon credit purchases and internal decarbonisation. Here, internal decarbonisation initiatives directly compete with spending on carbon credits. The second is the target effect, which occurs when companies allow emission offsetting to reach climate targets, offering a low-cost alternative to internal decarbonisation. However, the long-term impact of these effects on emission trajectories remains unclear. In practice, companies may increase their decarbonisation budgets once they realise that more funds are necessary. For instance, after committing USD 1 billion to decarbonisation in 2020, Delta Air Lines acknowledged in 2023 that additional resources would be required to achieve their emission reduction targets[47]. Moreover, access to emission offsetting could allow companies to achieve more ambitious climate targets than companies that do not use offsets for residual emissions.

The lack of significant correlation between the number of retired carbon credits and environmental performance suggests that voluntary carbon offsetting has historically not been associated with moral hazard. However, there is qualitative evidence that some oil and gas companies, like Shell, Eni, and Inpex, plan to integrate voluntary emission offsetting into their climate targets. Although we cannot assess whether these companies genuinely plan to reduce their emissions, this approach could result in moral hazard if companies purchase carbon credits that do not entail promised emission reductions rather than actively reduce their emissions. In such a scenario, the use of carbon credits could reflect the same type of moral hazard previously observed in the context of renewable energy attributes, where the claimed benefits fall short of real emission reductions. Lastly, some compliance pricing mechanisms, such as the Colombian and South African carbon taxes or the Korean and Californian ETSs, permit companies to use carbon credits sourced from voluntary carbon markets to meet part of their obligations[48]. Hence, if companies reduce

their compliance emission costs by retiring carbon credits, this can be associated with moral hazard.

Our findings entail several recommendations for policymakers. Voluntary emission offsetting is not associated with positive corporate environmental performance. Therefore, it is not a reliable alternative to regulatory measures, such as compliance carbon pricing. Policymakers should, therefore, focus on strengthening regulatory mechanisms to ensure substantial corporate contributions to emission reductions. Moreover, companies in emission-intensive sectors typically offset only small portions of their total scope 1, 2, and 3 emissions and allocate minimal funds to purchasing carbon credits. This highlights the importance of policies such as the European Union's directive to empower consumers for the green transition through better protection against unfair practices and through better information[49], which aims to prevent companies from making excessive environmental claims.

While we find clear indications that carbon credits play a minor role in corporate climate strategies, the study has potential limitations. First, we only consider a limited number of companies in three hard-to-abate sectors. Therefore, we might not identify small effect sizes of carbon credit usage on the change in scope 1 emissions and climate target ambition. However, few companies in hard-to-decarbonise sectors purchase substantial quantities of carbon credits and report on the details of their credit purchases and emissions. In hard-to-decarbonise sectors like cement, steel, and cargo shipping, no major company offset substantial shares of their emissions during the latest CDP reporting year[40]. Therefore, the effect of voluntary offsetting cannot be evaluated for those sectors. Second, we rely on self-reported company data, which are often criticised for their lack of credibility and comparability between companies. To limit the influence of reported data, we validated offsetting data with publicly available carbon credit registries and manually checked if outliers in our dataset were plausible.

Overall, the system of voluntary offsetting emissions is not associated with improved corporate sustainability performance. Most companies allocate minimal funds to voluntary carbon credits, which, even if redirected toward internal decarbonisation, would likely have little impact on their overall environmental performance. Further, the flexibility for companies to discontinue or alter their offsetting strategies at any time indicates that emission offsetting does not entail a solid commitment to decarbonising operations and value chains. Therefore, voluntary carbon offsetting should not be considered an indicator of superior corporate environmental performance. Instead, the public discourse on corporate decarbonisation should focus on progress in internal and value chain decarbonisation initiatives.

## Methods
### Quantitative data
Our main data source for numerical sustainability data is the CDP (formerly known as Carbon Disclosure Project) reporting cycle 2023. CDP has collected climate impact disclosure from companies since

2002. Scholars frequently use CDP data to study corporate climate performance over time[50–53]. In addition, we use financial data from S&P Capital IQ.

The study includes 89 oil and gas, airlines, and automobile manufacturing companies. These sectors are well-suited for investigating the role of voluntary emission offsetting, as they are characterised by high emissions and include some of the largest emission offsetting companies. In contrast, other hard-to-decarbonise industries, such as steel, cement, and maritime shipping, rarely engage in voluntary offsetting[40]. The sample includes all passenger airlines and automobile manufacturers that disclose their emissions to CDP. Further, we selected the 40 companies with the highest scope 1 emissions from the oil and gas sector due to the large number of companies in the CDP dataset. PJSC Lukoil was excluded from the analysis due to substantial structural changes in the Russian gas industry during the study period. Consequently, the final sample consists of 39 oil and gas companies, 27 passenger airlines, and 23 automobile manufacturers.

We use data from CDP's 2023 survey wave to obtain (1) companies' scope 1, 2 and 3 emissions, (2) carbon credit retirement data (number of credits and project types), (3) emission targets, and (4) purchased allowances under emission trading schemes. Companies' reporting years in the 2023 CDP survey wave end between 31. March 2022 and 31. March 2023, with the majority (70 companies) reporting for the calendar year 2022. In addition to the 2023 survey wave, we use CDP's 2018 survey wave to obtain historical scope 1 and 2 emission data. Since 36 companies in our sample did not participate in CDP's emission disclosure in 2018, we complement the data with information from corporate sustainability or annual reports. For nine companies (Koç Holding, San Miguel Corporation, Copa Holding, Kinder Morgan, Wizz Air Holding, Grupo Aeromexico, NFI Group, Hawaiian Holding, and Chorus Aviation), we have not found scope 1 emission data for 2018.

To improve data reliability, we cross-validate carbon credit retirement data with the largest voluntary carbon market registries of Verra, Gold Standard, and CDM. While the CDM was introduced to enable industrialised countries to reach their emission targets under the Kyoto Protocol, CDM credits can also be retired for voluntary purposes. If a company's cumulative credit retirements in those carbon market registries are larger than what the company reports to CDP, we use the registry data. Discrepancies between CDP data and registry data happen predominantly in cases where companies offset emissions for fossil products sold to clients (e.g. Inpex Corporation and PETRONAS).

## Historical emission performance

We define the historical emission performance as the ratio of scope 1 emissions reported in the CDP survey waves 2023 and 2018. We do not include scope 2 emissions since 20 companies in the sample did not report scope 2 emissions in 2017, and 13 further companies reported scope 2 emissions in 2017 without disclosing whether they used the location-based or market-based accounting approach. Further, location-based scope 2 emissions are heavily influenced by grid-emission factors, while market-based scope 2 emissions can be lowered through the purchase of renewable energy attributes that have been shown to have little effect on renewable energy expansion[28,29]. To ensure robustness, we present findings that include changes in combined scope 1 and 2 emissions in the supplementary information (Table S4). Further, we do not consider scope 3 emissions since current reporting practices do not allow for meaningful cross-organisational comparison[54]. We exclude Mercedes-Benz Group from the emission analysis due to the spin-out of Daimler Truck in December 2021. Further, we exclude Inpex Corporation from the emission analysis since it is an outlier with a 1085% increase in scope 1 over the study time. In the supplementary information we illustrate with a leave-one-out cross-validation that Inpex Corporation is the only data point that influences the results substantially (see Supplementary Information Fig. S1).

We perform ordinary least squares (OLS) regression using the python package statsmodels (version 0.14.1) to estimate the association between retired carbon credits and the change in historical emission reductions. We control for companies' sizes, industrial sectors, and continents of headquarters. We estimate:

$$Y_i = \beta_0 + \beta_1 X_i + \beta_2 C_i + \epsilon_i \tag{1}$$

where i indexes the companies, $Y_i$ is a company's ratio of scope 1 emissions reported in the CDP survey waves 2023 and 2018, $X_i$ is the number of carbon credits a company retired in CDP's 2023 reporting cycle, $C_i$ are the control variables, and $\epsilon_i$ represents the error term, capturing unobserved factors affecting $Y_i$. We control for revenue, sector, and continent of headquarters. We convert the categorical variables for the sector (automobile, oil and gas, and airlines) and the continent of headquarters (Asia, Europe, Latin America, and North America) into binary indicator columns (i.e. one-hot encoding). That means in the regression, each categorical variable equals 1 if a company belongs to a specific sector or is headquartered in a particular region and 0 otherwise. To avoid multicollinearity, we excluded the sectoral category "Airlines" and the geographical category "Asia" from the regression. These omitted categories serve as the reference groups against which the effects of other categories are compared.

## Climate target ambition

It is difficult to compare climate target ambition due to differences in scope, base years and target years. Therefore, corporate climate targets must be harmonised before they can be directly compared across companies[55]. Here we calculate the ambition for each emission scope (scope 1, 2, and 3) and then add the ambitions weighted with the relative importance of that scope for a specific industry (relative importance = avg. share of total emissions for scope n in industry X).

We use the following assumptions and simplifications to construct the target emission trajectories between 2020 and 2050 in line with previous studies that compared climate targets across organisations:

- Between intermediate targets, emission trajectories are linear[55].
- Emissions that are not covered by any target remain unchanged[55,56]
- We only consider company-wide targets, not product targets - e.g. when a company sells oil and gas but only has a target for their oil operation, we do not regard it since a reduction in oil emissions could be compensated by increased gas production. The only exception is when there are product-level targets for all main products of a company (e.g. separate targets for oil and gas operation). This assumption helps to avoid the difficulties of aggregating product-level emission-intensity targets for integrated energy companies that often sell different energy and non-energy products[56] by constructing a company-wide intensity metric.
- In line with SBTi's net zero standard[6], we accept both absolute and intensity targets. If, for the same year, intensity and absolute targets are given, we use the absolute target. While other studies also accept absolute and intensity targets[55,56], in contrast to Bolay et al. (2022), we do not use the associated expected change in absolute emissions for intensity targets that companies need to report to CDP. Instead, we directly use the targeted change in emission intensity to construct a company's emission trajectory. The deviation from previous studies is due to low data quality for the expected change in absolute emissions for intensity targets. Often, it is unclear if companies correctly use positive and negative numbers to indicate expected increases or decreases in absolute emissions. Further, it is not possible to verify the data since this data is typically not reported in annual or sustainability reports. To verify that companies that use intensity targets do not

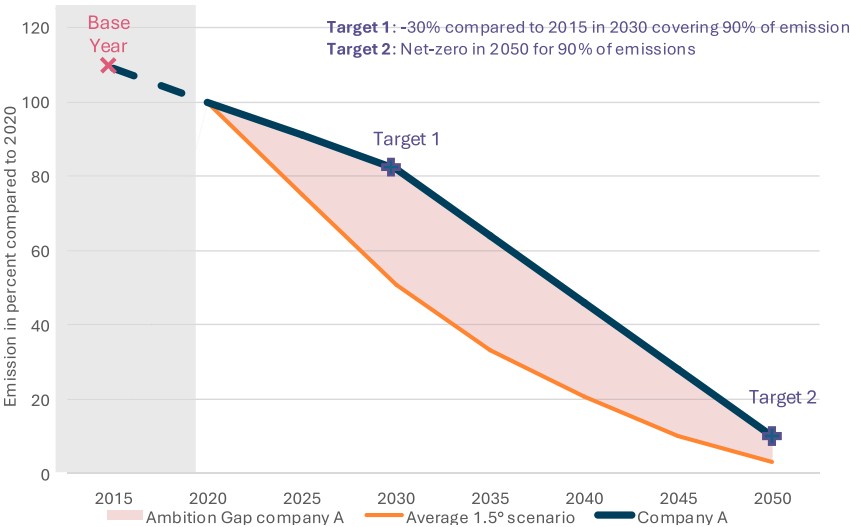

**Fig. 6 | Illustration of target ambition calculation.** Illustration of emission target quantification for company A. The red area is the magnitude of ambition. Since the targeted emission trajectory is higher than the 1.5° degree trajectory, the ambition is negative.

systematically set more ambitious climate targets, we show in the Supplementary Information that the share of a company's target that is covered by intensity targets is not significantly correlated with its target ambition (Table S5) and, therefore, does not bias the result reported in the regression analysis.

We assign an ambition score for each subtarget by comparing planned emission trajectories with emission reductions in line with the annual average of all 1.5 degree Celsius warming emission scenarios from IPCC's Sixth Assessment Report[57] Studies that evaluated climate targets compare target ambitions either to emission trajectories[58] or to targeted average annual change in emissions[55,56]. We choose to compare companies based on the targeted cumulative emissions until 2050 instead of targeted average annual changes in emissions since not only the average change in emissions but also the shape of the emission trajectory is directly linked to global warming. The choice of the reference emission trajectory is not relevant for the regression results since it is the same constant that is deducted from each observation and, hence, does not influence the regression coefficients. Further, we do not use different reference emission trajectories per sector or geography to avoid biasing our regression results. Instead, we explicitly control for geography and sector in the regression.

Figure 6 illustrates how we translate emission targets to emission trajectories and target ambition for the example company A. Company A's emission target only covers 90% of its emissions. Therefore, we assume that 10% of emissions remain unchanged. First, we construct the emission trajectory between the base year and target years, assuming emissions decline linearly. Second, we set 2020 as the reference year where emissions are at 100%. Finally, we quantify the target ambition as the area between the targeted company emission trajectory and the average of IPCC's 1.5 degree Celsius warming emission scenarios:

$$\text{Target Ambition} = \sum_{n=1}^{3}(\text{avg\_share\_scope}\_n$$
$$\times \left(\int_{2020}^{2050} \text{IPCC\_1.5\_scenario} \, dt \quad (2)\right.$$
$$\left. - \int_{2020}^{2050} \text{Company\_target\_scope}\_n \, dt\right))$$

Positive target ambition values indicate targets surpassing the reference scenario's ambition, negative target ambition values fall short of the reference scenario's ambition, and values of zero indicate an exact match with the average of IPCC's 1.5 degree Celsius warming emission scenarios.

We perform OLS using the python package statsmodels (version 0.14.1) to estimate the effect size of using carbon credits on the companies' climate target ambitions. We estimate:

$$Y_i = \beta_0 + \beta_1 X_i + \beta_2 C_i + \epsilon_i \quad (3)$$

where i indexes the companies, $Y_i$ is a company's climate target ambition (eq. (2)), $X_i$ is the number of carbon credits a company retired in CDP's 2023 reporting cycle, $C_i$ are the control variables, and $\epsilon_i$ represents the error term, capturing unobserved factors affecting $Y_i$. We control for revenue, sector, continent of headquarters, and share of emissions covered by intermediate targets on the path to net zero. We convert the categorical variables for the sector (automobile, oil and gas, and airlines) and the continent of headquarters (Asia, Europe, Latin America, and North America) into binary indicator columns (i.e. one-hot encoding). That means in the regression, each categorical variable equals 1 if a company belongs to a specific sector or is headquartered in a particular region and 0 otherwise. We include the share of emissions covered by intermediate targets (i.e. emission targets that are no net zero targets) to avoid systematically favouring companies that set only long-term net zero targets without intermediate goals. For instance, a company with a net zero target for 2050 but no intermediate targets would appear to have higher target ambition than the exemplary company depicted in Fig. 6, as the red area decreases without Target 1.

## Purpose of emission offsetting
Besides the quantitative evaluation of how many carbon credits companies retire, we qualitatively evaluate how companies use carbon credits over time using the qualitative data analysis software Atlas.ti. We manually scanned through 488 corporate sustainability and annual reports for the years 2014–2023 using the search words "offset", "carbon credit", "carbon market", "compensate", and "compensation". We assigned codes when the text passage revealed the purpose of voluntary carbon credit retirement. We tagged 522 text passages on carbon credit usage in 238 distinct documents.

We manually classify the use case of carbon credits into five categories:

- Retirement to meet emission targets: Companies plan to use carbon credits to reach an emission target. The quantity of retired carbon credits depends on the gap to target completion.
- Offsetting fixed portion of emissions: Companies define a fixed portion of their emissions to offset (e.g. scope 1, scope 2, specific product).
- Employees' business travel
- Customer Offsetting: Companies either offer customers the service to purchase carbon credits during checkout (mostly business to consumer) or offer pre-offset products (both business to consumer and business to business).
- Other: Offset usage that does not fit in other categories. These are often pilot projects to source carbon credits in preparation for compliance schemes or fixed-term marketing events.

## Carbon credit costs estimation

Prices companies pay for carbon credits are not publicly available at a granular level. Therefore, we estimate feasible price ranges by using the lowest prices reported by companies (USD 3.72 by easyJet[59]) in our sample as the low end of the price range and average credit prices 2022 by project type reported by Ecosystem Marketplace[60] as the upper end of the price range. Using Ecosystem Marketplace's data as an upper bound is a feasible assumption for the upper end of the feasible price range since companies in our sample are multinational companies purchasing relatively large quantities of carbon credits. Therefore, we assume these companies do not pay prices above the market average. We estimate the lower ($boundary_{low}$) and upper ($boundary_{high}$) boundary of the feasible carbon credit cost range for company $i$ purchasing carbon credits from project type $n$ as:

$$\text{budget boundary}_{low,i} = \text{credit price}_{easyJet} * \text{credits retired}_{CDP2023,i} \tag{4}$$

$$\text{budget boundary}_{high,i} = \sum_n \left( \text{avg credit price price}_n * \text{credits retired}_{n,i} \right) \tag{5}$$

To put the costs of carbon credits in perspective, we compare them with companies' CAPEX and costs of emission allowances under compliance emission trading schemes. CAPEX, which represents the funds a company spends to buy or improve assets, helps assess whether carbon credit costs are substantial enough to compete with investments in internal decarbonisation. Additionally, by comparing these costs to compliance emission pricing mechanisms, we evaluate whether carbon offsetting incentivises companies to accelerate decarbonisation efforts beyond existing regulation.

## ETS cost estimation

We estimate the costs for emission allowances in European emission trading schemes (EU-ETS, UK-ETS, and Switzerland-ETS) by multiplying the number of allowances a company purchased under a specific ETS with average allowance prices in 2022. Companies disclose the number of allocated allowances, purchased allowances, and overall emissions under ETSs in the CDP survey[40]. Average ETS prices in 2022 are based on the World Bank's carbon pricing database (EU ETS: USD 86.52, UK ETS: USD 98.99; Switzerland: 65.59 USD)[46]. The only exception is easyJet, which does not disclose the number of purchased allowances in the CDP report. Here we determine the number of purchased emission allowances as the difference between scope 1 emissions under the different ETSs disclosed to CDP[40] and publicly available information on allocated emission allowances[61–63].

## Reporting summary

Further information on research design is available in the Nature Portfolio Reporting Summary linked to this article.

## Data availability

All relevant data to reproduce plots can be found in our supplementary data and under https://github.com/n-stolz/nature_comms_negligible_role_carbon_offsetting.git. The data used in this article includes data points from CDP. The reproduction of any part of the CDP data by any third party is prohibited.

The data can be cited as: Niklas Stolz and Benedict Probst, The negligible role of carbon offsetting in corporate climate strategies, https://github.com/n-stolz/nature_comms_negligible_role_carbon_offsetting.git, https://doi.org/10.5281/zenodo.15634074, 2025.

## Code availability

All relevant code can be found under https://github.com/n-stolz/nature_comms_negligible_role_carbon_offsetting.git.

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

## Acknowledgements

Niklas Stolz is part of SPEED2ZERO, a Joint Initiative co-financed by the ETH Board. We would like to express our gratitude to Lambert Schneider, Malte Toetzke, and Volker Hoffmann for their valuable input and insightful discussions that contributed to the development of this work.

## Author contributions

N.S. designed and led the implementation of the study and collected and analysed the data. N.S. and B.S.P. developed the methodology and wrote the manuscript.

## Funding

## Competing interests

The authors declare no competing interests.
