## [Transparent Peer Review file · Nature Communications]

The negligible role of carbon offsetting in corporate climate strategies

Corresponding Author: Mr Niklas Stolz

Version 0:

Reviewer comments:

Reviewer #1

(Remarks to the Author)

This study sets out to understand to what extent companies' purchases of carbon credits impact their broader climate change strategy, for example, related to achieved emission reductions and future reduction targets. Although conflicting existing literature has both indicated (or hypothesized) a positive impact and a negative impact, the authors find no substantial impact, except potentially in the case of a handful of airlines (and other major purchasers of carbon credits). The research question is relevant and timely, the findings are interesting, the underlying methods appear appropriate and the study is well-written. I therefore support the publication of the study.

I mainly have minor comments:

- Page 1, abstract: consider replacing “we find no significant difference between companies that purchased credits and those that did not” by “we find no significant difference between the climate strategies of companies that purchased credits and those that did not.”
- Figure 1 and its captions: for someone like me that is not very familiar with the OSL regression, it would be nice if you could “hold the readers’s hand” and explain basic things, like: 1) “positive values indicate slower decarbonisation over the study time” compared to companies that do not buy credits? 2) “positive values indicate more ambitious targets” compared to companies that do not buy credits? 3) what is the meaning of the “sectoral categorical variables” and the “geographic categorical variables”? (I do not find the explanation in Methods) 4) Why is “number of retired credits” indicated in red? 5) what is the meaning of emission share in mid-term target? (I do not find the explanation in Methods).
- Page 8, line 9: with the wording “more meaningful decarbonization initiatives”, you appear to assume that carbon credits do not reduce emissions as efficiently as “investments in operations or value-chain decarbonisation”. However, from my understanding, your results do not say anything about, for example, the quantity of reduced emissions per dollar spent on carbon credits vs. “investments in operations or value-chain decarbonisation”? Check if an adjustment in wording is needed.
- Page 8, line 30: Discussion section: consider comparing your findings to broader literature on “moral hazard” of market-based instruments in carbon accounting, for example “Ascui, F., Brander, M., Cojoianu, T. and Li, Q., 2021. Moral hazard and the market-based method: Does using REAs affect corporate emissions performance?. In *Academy of Management Proceedings* (Vol. 2021, No. 1, p. 15686). Briarcliff Manor, NY 10510: Academy of Management.”.
- Page 10, line 29: the section on “Climate Target Ambition” is clearly written and seems methodologically sound. However, I encourage the authors to reference existing literature that also has the purpose of establishing comparability between the emission reduction targets of different companies, potentially in the light of the Paris Agreement temperature ceiling, and then highlight and justify any differences that your approach might have. Here are a handful of key studies that I am aware of:
 - o Dietz, S., Gardiner, D., Jahn, V. and Noels, J., 2023. Carbon Performance assessment of oil & gas producers: note on methodology. Transition Pathway Initiative Centre, London School of Economics and Political Science.
 - o Bolay, A.F., Bjørn, A., Weber, O. and Margni, M., 2022. Prospective sectoral GHG benchmarks based on corporate climate mitigation targets. *Journal of Cleaner Production*, 376, p.134220.
 - o Bjørn, A., Lloyd, S. and Matthews, D., 2021. From the Paris Agreement to corporate climate commitments: evaluation of seven methods for setting ‘science-based’ emission targets. *Environmental Research Letters*, 16(5), p.054019.

(Remarks on code availability)

Reviewer #2

(Remarks to the Author)

Manuscript Title: The negligible role of carbon offsetting in corporate climate strategies

I express sincere gratitude to the editorial board for offering to review the captioned paper. The manuscript caters to a critical and timely research topic and presents robust and useful findings. The results suggest that carbon credit retirement does not affect the climate strategy of the sample firms as well as the internal decarbonization (in most cases). Provided the quality of the work, I would recommend authors consider the below points to enrich the manuscript value.

- The introduction section fairly describes the need for this research. Yet, the authors should consider adding a paragraph of review of relevant past studies to further strengthen the contributions to the body of knowledge.
- Under the Methods section, the author should describe the sample selection process leading to selection of 89 firms.
- The findings have huge potential to provide critical policy implications for regulators. Authors should incorporate a section for the same and discuss.

(Remarks on code availability)

NA

Version 1:

Reviewer comments:

Reviewer #1

(Remarks to the Author)

The authors have done an excellent job at responding to my original review comments, and the comments of the other reviewer. The paper has been substantially improved as a result and I have no further comments.

(Remarks on code availability)

Reviewer #2

(Remarks to the Author)

I have reviewed the revised manuscript and found that authors have sufficiently addressed the comments provided earlier. I recommend the article for publication.

(Remarks on code availability)

NA

Point-by-point response to the referees

Dear referees,

Thank you for your constructive reviews on our submitted manuscript, "The negligible role of carbon offsetting in corporate climate strategies". We welcome the opportunity to submit a thoroughly revised version of our manuscript together with this point-by-point revision letter.

We highly appreciate the extremely helpful feedback from the reviewers. In the following, we would like to summarise the most important changes that we incorporated during the revision. We are confident that these changes address all raised concerns. In addition, we strongly believe that they led to significant improvements.

- 1. Refinement of integration in literature:** As suggested by reviewer 2, we have improved the integration of our study into the existing literature. To achieve this, we added a paragraph in the Introduction that highlights key findings from previous studies on the relationship between corporate carbon management practices and environmental performance. This addition strengthens the foundation of our research and clarifies its contribution to the field.
- 2. Discussion on moral hazard:** Based on the suggestion of reviewer 1, we have refined our study to include a discussion around moral hazard associated with emission offsetting. To address this, we introduced the concept of moral hazard and relevant findings for other market-based carbon accounting instruments to the Introduction. Additionally, we incorporated a dedicated section in the Discussion that examines the potential risks associated with moral hazard.
- 3. Improved robustness of Results:** We ran additional tests to ensure the robustness of our regression results. After evaluating the influence of single observations on the regression analysis (Figure 1), we decided to exclude INPEX Corporation from the emission analysis and to re-run the regression on the change in scope 1 emissions (Fig. 1a) since it is an outlier that strongly influences the results. The conclusions of the analysis remain the same, and the messages are reinforced by additional robustness checks presented in the Supplementary Information section S1.
- 4. Refinement of Methodology section:** Based on the suggestions from Reviewers 1 and 2, we have refined the methodology section to improve its clarity and better integrate it with the existing literature. Specifically, we added a section detailing the selection of companies for the study and enhanced the section on quantifying corporate climate targets to make the connections to previous studies more explicit. Additionally, we revised the caption of Figure 1 to provide clearer guidance for readers.
- 5. Recommendation for policymakers:** As suggested by Reviewer 2 we added a section on recommendations for policymakers to the discussion. We highlight that policymakers should not view voluntary carbon markets as an alternative to compliance mechanisms. Further, our work strengthens the rationale of policies like the European Union's Green Claims Directive that prevent companies from making excessive environmental claims

based on emission offsetting. Finally, we warn policymakers from expanding the use-case of voluntary carbon credits without improving market mechanisms.

We believe our work on the role of carbon credits in corporate climate strategies is highly relevant to ensure effective Net Zero transitions. Given the prominent role of carbon credits in the public discourse on decarbonisation and some companies' communication on their climate strategies, we believe that our study will help to guide the debate on voluntary emission offsetting, which stakeholders from the industry currently dominate.

Based on the reviewers' feedback, we have substantially improved the transparency of the method, the integration with the literature, and the implications of our findings. We are confident that the manuscript has improved considerably.

Reviewer 1:

Reviewer Comment:

This study sets out to understand to what extent companies' purchases of carbon credits impact their broader climate change strategy, for example, related to achieved emission reductions and future reduction targets. Although conflicting existing literature has both indicated (or hypothesized) a positive impact and a negative impact, the authors find no substantial impact, except potentially in the case of a handful of airlines (and other major purchasers of carbon credits). The research question is relevant and timely, the findings are interesting, the underlying methods appear appropriate and the study is well-written. I therefore support the publication of the study.

Author Response:

We thank the reviewer for the positive and constructive feedback. We indicate additions to the manuscript in **blue** and deleted text in **red**.

Reviewer Comment 1:

- Page 1, abstract: consider replacing “we find no significant difference between companies that purchased credits and those that did not” by “we find no significant difference between the climate strategies of companies that purchased credits and those that did not.”

Author Response:

We thank the reviewer for this suggestion. We changed the abstract accordingly (p. 1, l.9).

Reviewer Comment 2:

- Figure 1 and its captions: for someone like me that is not very familiar with the OSL regression, it would be nice if you could “hold the readers's hand” and explain basic things, like:
1) “positive values indicate slower decarbonisation over the study time” compared to companies

that do not buy credits? 2) “positive values indicate more ambitious targets” compared to companies that do not buy credits? 3) what is the meaning of the “sectoral categorical variables” and the “geographic categorical variables”? (I do not find the explanation in Methods) 4) Why is “number of retired credits” indicated in red? 5) what is the meaning of emission share in mid-term target? (I do not find the explanation in Methods).

Author Response:

We thank the reviewer for this helpful comment. We added clarification to the caption of Figure 1 (p. 3) and the Methods section (p. 11, l. 1-5; p. 12 l. 5-11). Additionally, we provide a brief explanation of the unclear points:

(1) & (2): The regression coefficients reflect not only the number of carbon credits a company retired but also the influence of all other explanatory variables in the model. These coefficients indicate whether explanatory variables are significantly correlated with the outcome variables (a) change in scope 1 emissions over time and (b) climate target ambition. The coefficients do not compare companies that retire carbon credits to those that do not. Instead, they show whether an increase in the number of carbon credits retired (starting from zero for companies that do not retire credits) is significantly associated with the decarbonisation speed and the ambition of climate targets. We now make this explicit in the caption of Figure 1 by stating (p.3): “In (a), positive regression coefficients indicate a negative relationship between the explanatory variables (on y-axis) and decarbonisation speed (x-axis), suggesting that as the explanatory variables increase, we observe a decreased decarbonisation speed. In (b), positive regression coefficients indicate a positive relationship between the explanatory variables (y-axis) and climate target ambition (x-axis), suggesting that as the explanatory variables increase, we observe an increased climate target ambition. The sectoral categorical variables are relative to the aviation sector, and the geographic categorical variables are relative to headquarters in Asia.”

(3): We clarified the meaning of sectoral variables in the Methods section (p. 11, l.4-9):

“We convert the categorical variables for the sector (automobile, oil and gas, and airlines) and the continent of headquarters (Asia, Europe, Latin America, and North America) into binary indicator columns (i.e. one-hot encoding). That means in the regression, each categorical variable equals 1 if a company belongs to a specific sector or is headquartered in a particular region and 0 otherwise. To avoid multicollinearity, we excluded the sectoral category "Airlines" and the geographical category "Asia" from the regression. These omitted categories serve as the reference groups against which the effects of other categories are compared.”

(4): We highlighted the number of retired carbon credits to help the reader identify the outcome variable of interest for the study. To avoid confusion, we changed the colour of the errorbar to blue and explicitly stated in the caption of Figure 1 (p. 3): “The label of retired carbon credits is written in bold as it represents the primary outcome variable of interest in the study.”

(5): We clarified the reason for including the share of intermediate targets in the Methods section (p. 12, l. 10-14):

“We include the share of emissions covered by intermediate targets (i.e. emission targets that are no net zero targets) to avoid systematically favouring companies that set only long-term net zero targets without intermediate goals. For instance, a company with a net zero target for 2050 but no intermediate targets would appear to have higher target ambition than the exemplary company depicted in Figure 6, as the red area decreases without Target 1..”

Reviewer Comment 3:

• Page 8, line 9: with the wording “more meaningful decarbonization initiatives”, you appear to assume that carbon credits do not reduce emissions as efficiently as “investments in operations or value-chain decarbonisation”. However, from my understanding, your results do not say anything about, for example, the quantity of reduced emissions per dollar spent on carbon credits vs. “investments in operations or value-chain decarbonisation”? Check if an adjustment in wording is needed.

Author Response:

We thank the reviewer for this valuable input. We changed the wording according to the reviewer's suggestion (p.8, l. 9) to “investments in operations or value chain decarbonisation”. Initially, we based our wording on recent findings by Probst et al. (2024) that less than 16% of carbon credits constitute real emission reductions. However, we do not evaluate this in our study, and based on our data and analysis, as you rightfully remarked, we cannot conclude that investments in internal and value-chain decarbonisation are more effective at reducing emissions per dollar spent than carbon credits.

Reviewer Comment 4:

• Page 8, line 30: Discussion section: consider comparing your findings to broader literature on “moral hazard” of market-based instruments in carbon accounting, for example “Ascui, F., Brander, M., Cojoianu, T. and Li, Q., 2021. Moral hazard and the market-based method: Does using REAs affect corporate emissions performance?. In Academy of Management Proceedings (Vol. 2021, No. 1, p. 15686). Briarcliff Manor, NY 10510: Academy of Management.”.

Author Response:

We thank the reviewer for this valuable addition. We added a section to introduce the moral hazard concept (p.1, l. 24-37) and the literature on the moral hazard of renewable energy certificates (p. 2, l. 9-15) to the Introduction. Further, we added a section about how moral hazard of carbon credit usage is linked to renewable energy certificates to the Discussion (p. 9, l. 43-51).

We now write in the introduction (p.1, l. 34-38): “Purchasing carbon credits instead of pursuing – potentially more effective – internal decarbonisation can be conceptualised as a moral hazard. Moral hazard occurs when actors take on higher risks or engage in socially suboptimal behaviour because they are shielded from the consequences of their actions (Baker, 1996) . Therefore, emission offsetting might lead to moral hazard when companies neglect internal and value chain emission reductions because the improved public perception achieved through emission offsetting shields them from the risk of reputational damage, public scrutiny, or governmental regulation.” and (p. 2., l.10-17): “While research on companies' use of carbon credits is still nascent, research on renewable energy attributes (REAs), another market-based carbon accounting tool, is more advanced. REAs allow companies to verify and claim the purchase of renewable energy, directly reducing market-based scope 2 emissions under the Greenhouse Gas Protocol and Science-Based Targets initiative (SBTi)(*The Greenhouse Gas Protocol*, 2015) . Unlike voluntary carbon credits, REAs can be counted towards SBTi goals. Ascui et al. (Ascui et al., 2021) show that companies using REAs tend to increase their scope 1 and 2 emissions without improving energy efficiency compared to peers who do not use them, which indicates their potential to induce moral hazard (Ascui et al., 2021). Additionally, setting targets for market-based scope 2 emissions and achieving them by purchasing REAs might undermine the integrity of SBTi because these certificates do not lead to real emission reductions (Bjørn et al., 2022; Brander et al., 2018; Gillenwater et al., 2014).”

In the Discussion, we added a paragraph on potential moral hazard, linking it to REAs. We now write (p.9, l.30-39): “The lack of significant correlation between the number of retired carbon credits and environmental performance suggests that voluntary carbon offsetting has historically not been associated with moral hazard. However, there is qualitative evidence that some oil and gas companies, like Shell, Eni, and Inpex, plan to integrate voluntary emission offsetting into their climate targets. This approach could result in moral hazard if companies purchase carbon credits that do not entail promised emission reductions rather than actively reduce their emissions. In such a scenario, the use of carbon credits could reflect the same type of moral hazard previously observed in the context of renewable energy attributes, where the claimed benefits fall short of real emission reductions. Lastly, some compliance pricing mechanisms, such as the Colombian and South African carbon taxes or the Korean and Californian ETSs, permit companies to use carbon credits sourced from voluntary carbon markets to meet part of their obligations (UNFCCC, 2023). Hence, if companies reduce their compliance emission costs by retiring carbon credits, this can be associated with moral hazard.”

Reviewer Comment 5:

- Page 10, line 29: the section on “Climate Target Ambition” is clearly written and seems methodologically sound. However, I encourage the authors to reference existing literature that also has the purpose of establishing comparability between the emission reduction targets of

different companies, potentially in the light of the Paris Agreement temperature ceiling, and then highlight and justify any differences that your approach might have. Here are a handful of key studies that I am aware of:

- o Dietz, S., Gardiner, D., Jahn, V. and Noels, J., 2023. Carbon Performance assessment of oil & gas producers: note on methodology. Transition Pathway Initiative Centre, London School of Economics and Political Science.
- o Bolay, A.F., Bjørn, A., Weber, O. and Margni, M., 2022. Prospective sectoral GHG benchmarks based on corporate climate mitigation targets. *Journal of Cleaner Production*, 376, p.134220.
- o Bjørn, A., Lloyd, S. and Matthews, D., 2021. From the Paris Agreement to corporate climate commitments: evaluation of seven methods for setting 'science-based' emission targets. *Environmental Research Letters*, 16(5), p.054019.

Author Response:

We thank the reviewer for raising this critical point. Our approach is very similar to those used in the studies listed by the reviewer. We significantly revised the subsection "Climate Target Ambition" of the Methods to more clearly discuss the similarities and differences between the approaches..

The main difference between the approaches presented in the proposed literature and our analysis is how intensity targets are handled. When companies reported an intensity target in the CDP survey, Bjørn et al. (2021) used the expected change in absolute emissions that companies must report. However, we decided against using this metric since the data quality of this survey field is low, and some large companies like Shell did not report this number. The low data quality is especially concerning since the expected change in absolute emissions is not usually disclosed in other sources like websites or sustainability reports and can not be verified. Consequently, we decided to treat absolute and intensity targets equally and add a sensitivity check to the Supplementary Information (Table S5) to ensure the presence of intensity targets does not alter our main results (Table S6, p. 21). The sensitivity check shows that the presence of an intensity target is not significantly correlated with companies' climate target ambitions.

The methodology section on quantifying climate target ambition now reads (p. 11, l. 11-44):

"It is difficult to compare climate target ambition due to differences in scope, base years and target years. Therefore, corporate climate targets must be harmonised before they can be directly compared across companies (Bolay et al., 2022). Here we calculate the ambition for each emission scope (scope 1, 2, and 3) and then add the ambitions weighted with the relative importance of that scope for a specific industry (relative importance = avg. share of total emissions for scope n in industry X).

We use the following assumptions and simplifications to construct the target emission trajectories between 2020 and 2050 in line with previous studies that compared climate targets across organisations:

- Between intermediate targets, emission trajectories are linear (Bolay et al., 2022)
- Emissions that are not covered by any target remain unchanged (Bolay et al., 2022; Simon Dietz et al., 2021)
- We only consider company-wide targets, not product targets - e.g. when a company sells oil and gas but only has a target for their oil operation, we do not regard it since a reduction in oil emissions could be compensated by increased gas production. The only exception is when there are product-level targets for all main products of a company (e.g. separate targets for oil and gas operation). This assumption helps to avoid the difficulties of aggregating product-level emission-intensity targets for integrated energy companies that often sell different energy and non-energy products (Simon Dietz et al., 2021) by constructing a company-wide intensity metric.
- In line with SBTi's net zero standard (SBTi, 2023), we accept both absolute and intensity targets. If, for the same year, intensity and absolute targets are given, we use the absolute target. While other studies also accept absolute and intensity targets (Bolay et al., 2022; Simon Dietz et al., 2021), in contrast to Bolay et al. (2022), we do not use the associated expected change in absolute emissions for intensity targets that companies need to report to CDP. Instead, we directly use the targeted change in emission intensity to construct a company's emission trajectory. The deviation from previous studies is due to low data quality for the expected change in absolute emissions for intensity targets. Often, it is unclear if companies correctly use positive and negative numbers to indicate expected increases or decreases in absolute emissions. Further, it is not possible to verify the data since this data is typically not reported in annual or sustainability reports. To verify that companies that use intensity targets do not systematically set more ambitious climate targets, we show in the Supplementary Information that the share of a company's target that is covered by intensity targets is not significantly correlated with its target ambition (Tab. S5) and, therefore, does not bias the result reported in the regression analysis.

We assign an ambition score for each subtarget by comparing planned emission trajectories with emission reductions in line with the annual average of all 1.5 degree Celsius warming emission scenarios from IPCC's Sixth Assessment Report (Byers et al., 2022). Studies that evaluated climate targets compare target ambitions either to emission trajectories (Bjørn et al., 2021) or to targeted average annual change in emissions (Bolay et al., 2022; Simon Dietz et al., 2021). We choose to compare companies based on the targeted cumulative emissions until 2050 instead of targeted average annual changes in emissions since not only the average change in emissions but also the shape of the emission trajectory is directly linked to global warming. The choice of the reference emission trajectory is not relevant for the regression results since it is the same constant that is deducted from each observation and, hence, does not influence the regression coefficients. Further, we do not use different reference emission trajectories per sector or geography to avoid biasing our regression results. Instead, we explicitly control for geography and sector in the regression.

Reviewer #2 (Remarks to the Author):

Manuscript Title: The negligible role of carbon offsetting in corporate climate strategies

Reviewer Comment:

I express sincere gratitude to the editorial board for offering to review the captioned paper. The manuscript caters to a critical and timely research topic and presents robust and useful findings. The results suggest that carbon credit retirement does not affect the climate strategy of the sample firms as well as the internal decarbonization (in most cases). Provided the quality of the work, I would recommend authors consider the below points to enrich the manuscript value.

Author Response:

We thank the reviewer for the very positive and constructive feedback.

Reviewer Comment:

- The introduction section fairly describes the need for this research. Yet, the authors should consider adding a paragraph of review of relevant past studies to further strengthen the contributions to the body of knowledge.

Author Response:

We thank the reviewer for this constructive feedback. We added a paragraph on past studies on the relationship between corporate carbon management and emission performance and evidence around the influence of renewable energy certificates, another market-based carbon accounting tool, on corporate climate performance (p. 2, l. 1-17). We now write:

“There is mixed evidence regarding the relationship between corporate carbon management practices and subsequent emission reductions. While some studies demonstrate a positive relationship between corporate carbon disclosure and emission reductions (Downar et al., 2021; Qian & Schaltegger, 2017), others find this link only among emission-intensive companies (Hsueh, 2019). Conversely, to our knowledge, no study to date has established a significant relationship between adopting reporting guidelines, such as the Global Reporting Initiative (GRI) and improved corporate emission performance (Belkhir et al., 2017; Haque & Ntim, 2020). Further, the impact of corporate climate strategies on emission reductions remains ambiguous. For example, there is no relationship between the mere presence of corporate climate targets and subsequent decarbonisation (Dahlmann et al., 2019), though more ambitious targets are associated with greater emission reductions (Dahlmann et al., 2019; Ioannou et al., 2016). Recent findings suggest that only a comprehensive mix of corporate climate instruments (e.g., absolute emission targets, internal carbon prices, value chain engagement) is linked to absolute emission reductions (Klaaßen & Stoll, 2021)

While research on companies' use of carbon credits is still nascent, research on renewable energy attributes (REAs), another market-based carbon accounting tool, is more advanced. REAs allow companies to verify and claim the purchase of renewable energy, directly reducing market-based scope 2 emissions under the Greenhouse Gas Protocol and Science-Based Targets initiative (SBTi) (*The Greenhouse Gas Protocol*, 2015). Unlike voluntary carbon credits, REAs can be counted towards SBTi goals. Ascui et al. show that companies using REAs tend to increase their scope 1 and 2 emissions without improving energy efficiency compared to peers who do not use them, which indicates their potential to induce moral hazard (Ascui et al., 2021). Additionally, setting targets for market-based scope 2 emissions and achieving them by purchasing REAs might undermine the integrity of SBTi because these certificates do not lead to real emission reductions (Bjørn et al., 2022; Brander et al., 2018; Gillenwater et al., 2014).”

Reviewer Comment:

- Under the Methods section, the author should describe the sample selection process leading to selection of 89 firms.

Author Response:

We thank the reviewer for this important clarification question. We added a paragraph to the methods outlining the sample selection (p. 10, l. 13-20):

“The study includes 89 oil and gas, airlines, and automobile manufacturing companies. These sectors are well-suited for investigating the role of voluntary emission offsetting, as they are characterised by high emissions and include some of the largest emission offsetting companies. In contrast, other hard-to-decarbonise industries, such as steel, cement, and maritime shipping,

rarely engage in voluntary offsetting (CDP, 2023). The sample includes all passenger airlines and automobile manufacturers that disclose their emissions to CDP. We selected the 40 companies with the highest scope 1 emissions for the oil and gas sector due to the large number of companies in the CDP dataset. PJSC Lukoil was excluded from the analysis due to significant structural changes in the Russian gas industry during the study period. Consequently, the final sample consists of 39 oil and gas companies, 27 passenger airlines, and 23 automobile manufacturers.”

Reviewer Comment:

- The findings have huge potential to provide critical policy implications for regulators. Authors should incorporate a section for the same and discuss.

Author Response:

We thank the reviewer for this valuable addition. We have added policy recommendations to the Discussion (p. 9, l. 40-46). We now state:

“Our findings entail several recommendations for policymakers. Voluntary emission offsetting is not associated with positive corporate environmental performance or high costs for companies. Therefore, it is not a reliable alternative to regulatory measures, such as compliance carbon pricing. Policymakers should, therefore, focus on strengthening regulatory mechanisms to ensure substantial corporate contributions to emission reductions. Moreover, companies in emission-intensive sectors typically offset only small portions of their total Scope 1, 2, and 3 emissions and allocate minimal funds to purchasing carbon credits. This highlights the importance of policies such as the European Union's Green Claims Directive (Green Claims Directive, 2023), which aims to prevent companies from making excessive environmental claims.”

References

- Ascui, F., Brander, M., Cojoianu, T., & Li, Q. (2021). Moral hazard and the market-based method: Does using REAs affect corporate emissions performance? *Academy of Management Proceedings*, 2021(1), 15686. <https://doi.org/10.5465/AMBPP.2021.15686abstract>
- Baker, T. (1996). On the Genealogy of Moral Hazard. *Texas Law Review*, 75(2), 237–292.
- Belkhir, L., Bernard, S., & Abdelgadir, S. (2017). Does GRI reporting impact environmental sustainability? A cross-industry analysis of CO2 emissions performance between GRI-reporting and non-reporting companies. *Management of Environmental Quality: An International Journal*, 28(2), 138–155. <https://doi.org/10.1108/MEQ-10-2015-0191>
- Bjørn, A., Lloyd, S. M., Brander, M., & Matthews, H. D. (2022). Renewable energy certificates threaten the integrity of corporate science-based targets. *Nature Climate Change*, 12(6), 539–546. <https://doi.org/10.1038/s41558-022-01379-5>
- Bjørn, A., Lloyd, S., & Matthews, D. (2021). From the Paris Agreement to corporate climate commitments: Evaluation of seven methods for setting ‘science-based’ emission targets. *Environmental Research Letters*, 16(5), 054019. <https://doi.org/10.1088/1748-9326/abe57b>
- Bolay, A.-F., Bjørn, A., Weber, O., & Margni, M. (2022). Prospective sectoral GHG benchmarks based on corporate climate mitigation targets. *Journal of Cleaner Production*, 376, 134220. <https://doi.org/10.1016/j.jclepro.2022.134220>
- Brander, M., Gillenwater, M., & Ascui, F. (2018). Creative accounting: A critical perspective on the market-based method for reporting purchased electricity (scope 2) emissions. *Energy Policy*, 112, 29–33. <https://doi.org/10.1016/j.enpol.2017.09.051>
- Byers, E., Krey, V., Kriegler, E., Riahi, K., Schaeffer, R., Kikstra, J., Lamboll, R., Nicholls, Z., Sandstad, M., Smith, C., van der Wijst, K., Al -Khourdajie, A., Lecocq, F., Portugal-Pereira, J., Saheb, Y., Stromman, A., Winkler, H., Auer, C., Brutschin, E., ... van Vuuren, D. (2022). *AR6 Scenarios Database* (Version 1.1) [Dataset]. Intergovernmental Panel on Climate Change. <https://doi.org/10.5281/zenodo.7197970>
- Dahlmann, F., Branicki, L., & Brammer, S. (2019). Managing Carbon Aspirations: The Influence of Corporate Climate Change Targets on Environmental Performance. *Journal of Business Ethics*, 158(1), 1–24. <https://doi.org/10.1007/s10551-017-3731-z>
- Downar, B., Ernstberger, J., Reichelstein, S., Schwenen, S., & Zaklan, A. (2021). The impact of carbon disclosure mandates on emissions and financial operating performance. *Review of Accounting Studies*, 26(3), 1137–1175. <https://doi.org/10.1007/s11142-021-09611-x>
- Gillenwater, M., Lu, X., & Fischlein, M. (2014). Additionality of wind energy investments in the U.S. voluntary green power market. *Renewable Energy*, 63, 452–457. <https://doi.org/10.1016/j.renene.2013.10.003>
- Haque, F., & Ntim, C. G. (2020). Executive Compensation, Sustainable Compensation Policy, Carbon Performance and Market Value. *British Journal of Management*, 31(3), 525–546. <https://doi.org/10.1111/1467-8551.12395>
- Hsueh, L. (2019). Voluntary climate action and credible regulatory threat: Evidence from the carbon disclosure project. *Journal of Regulatory Economics*, 56(2), 188–225. <https://doi.org/10.1007/s11149-019-09390-z>

- Ioannou, I., Li, S. X., & Serafeim, G. (2016). The Effect of Target Difficulty on Target Completion: The Case of Reducing Carbon Emissions. *The Accounting Review*, 91(5), 1467–1492. <https://doi.org/10.2308/accr-51307>
- Klaaßen, L., & Stoll, C. (2021). Harmonizing corporate carbon footprints. *Nature Communications*, 12(1), Article 1. <https://doi.org/10.1038/s41467-021-26349-x>
- Proposal for a DIRECTIVE OF THE EUROPEAN PARLIAMENT AND OF THE COUNCIL on Substantiation and Communication of Explicit Environmental Claims (Green Claims Directive), No. COM(2023) 166, European Commission (2023).
- Qian, W., & Schaltegger, S. (2017). Revisiting carbon disclosure and performance: Legitimacy and management views. *The British Accounting Review*, 49(4), 365–379. <https://doi.org/10.1016/j.bar.2017.05.005>
- SBTi. (2023). *SBTi Corporate Net-Zero Standard v1.1* (Version 1.1).
- Simon Dietz, Dan Gardiner, Nikolaus Hastreiter, Valentin Jahn, & Jolien Noels. (2021). Carbon Performance assessment of oil & gas producers: Note on methodology. *Transition Pathway Initiative*. <https://transitionpathwayinitiative.org/publications/uploads/2021-carbon-performance-assessment-of-oil-gas-producers-note-on-methodology>
- The Greenhouse Gas Protocol—A Corporate Accounting and Reporting Standard*. (2015).
- UNFCCC. (2023). *Regional Climate Week Asia-Pacific—The role of carbon credits*. https://unfccc.int/sites/default/files/resource/Session%201d_The%20role%20of%20carbon%20credits.pdf